# Task-Oriented Thigh Ultrasound Phenotyping via nnU-Net: From Segmentation to Biomarkers

**Mara Concepción** [*1,5]                    MARA.CONCEPCION@UNAVARRA.ES
**Paula Crespo**[*1]                    CRESPO.154323@E.UNAVARRA.ES
**Rafael Cabeza**[1,2]                    RCABEZA@UNAVARRA.ES
**Fernando Idoate**[3,4]                    FERNANDO.IDOATE@MUTUANAVARRA.ES
**Idoia Labayen**[4,5,6,7]                    IDOIA.LABAYEN@UNAVARRA.ES
**Arantxa Villanueva**[1,2]                    AVILLA@UNAVARRA.ES

[1] *Department of Electrical, Electronic and Communications Engineering, Public University of Navarra, Spain*

[2] *Institute of Smart Cities (ISC), Public University of Navarra, Spain*

[3] *Radiology Department, Mutua Navarra, Spain*

[4] *Department of Health Sciences, Public University of Navarra, Spain*

[5] *Institute for Sustainability and Food Chain Innovation (IS-FOOD), Public University of Navarra, Spain*

[6] *Navarra Institute for Health Research (IdiSNA), Pamplona, Spain*

[7] *Centro de Investigación Biomédica en Red – Physiopathology of Obesity and Nutrition (CIBER-OBN), Instituto de Salud Carlos III, 28029 Madrid, Spain*

## Abstract

While accurate assessment of muscle and adipose tissue is essential for studying metabolic disorders, current ultrasound (US) approaches are often limited by operator dependency and poor reproducibility. We present a fully automatic pipeline for the segmentation and quantitative analysis of thigh US images across clinical cohorts with varying body composition profiles. Using a self-configuring nnU-Net, we perform multi-class segmentation of skin, rectus femoris, and femur, enabling the automated extraction of morphometric biomarkers and muscle radiomics. Beyond conventional spatial overlap metrics, we implement a task-oriented validation on an independent test set by evaluating the agreement between expert-derived and predicted clinical features. Our results demonstrate excellent biomarker agreement (CCC > 0.95), suggesting the potential of this approach for high-throughput, automated ultrasound-based phenotyping in clinical research.

**Keywords:** Ultrasound, medical image segmentation, nnU-Net, radiomics, muscle analysis

## 1. Introduction

Traditional anthropometric measures for metabolic assessment fail to capture the complexity of body composition, potentially masking high-risk phenotypes such as sarcopenic obesity (Mouchti et al., 2023; Christakoudi et al., 2022). Imaging-derived biomarkers provide a more precise characterization of adipose tissue distribution and muscle mass composition, which are strongly associated with metabolic alterations (Tarabeih et al., 2024; Bibi et al.,

---

[*] Contributed equally

2023). Ultrasound (US) offers a portable, less expensive, and non-invasive alternative to CT and MRI (Prado et al., 2026), but its clinical reliability is often limited by operator-dependent measurements (Bell et al., 2024). Moreover, the noise and low contrast make manual or classical automated segmentation unreliable (Sharma et al., 2026).

Deep learning, particularly U-Net-based architectures, has significantly improved medical image segmentation (Litjens et al., 2017; Ronneberger et al., 2015). However, their performance depends on extensive manual tuning, limiting generalization across datasets (Isensee et al., 2021). The nnU-Net framework addresses this limitation through a self-configuring pipeline and has shown strong performance in musculoskeletal ultrasound segmentation tasks (Zhang et al., 2024). Nevertheless, most approaches focus on overlap-based metrics such as the Dice Similarity Coefficient (DSC), without assessing whether the predicted segmentations are suitable for reliable biomarker extraction.

In this work, we propose a fully automatic pipeline for multi-class segmentation (skin, rectus femoris (RF), and femur) in ultrasound images across clinical cohorts using nnU-Net. The proposed framework enables the extraction of morphometric biomarkers—including subcutaneous adipose tissue (SAT), RF, and vastus intermedius (VI) thickness—as well as muscle radiomic texture features. These biomarkers have previously shown strong agreement with MRI-derived measurements and significant associations with clinically relevant functional and metabolic outcomes (Mechelli et al., 2019; Wilkinson et al., 2021; Bell et al., 2022). Furthermore, we perform a task-oriented validation by evaluating the agreement between biomarkers derived from ground-truth and predicted segmentations, demonstrating the reliability of the proposed approach for quantitative clinical analysis.

## 2. Methods

**Dataset and Ground Truth Generation.** The dataset comprises 499 US series from 143 participants across three clinical cohorts, including two clinical trials (NCT05310721, NCT05912309) and a normal-weight group (Dote-Montero et al., 2024; Alfaro-Magallanes et al., 2025). Ground truth masks for skin and RF were generated from expert annotations. For the femur, point-based coordinates were used to construct polygonal bounding boxes, which were then converted into binary masks.

**Training Strategy.** A 2D nnU-Net model was trained using its self-configuring default settings. The dataset was split into a training/validation pool and an independent test set ($n = 71$), ensuring patient-level separation to prevent data leakage. Model performance was estimated using 5-fold cross-validation with predefined splits to ensure a balanced representation of each cohort and evaluated on the unseen test set.

**Biomarker Extraction and Evaluation.** From the predicted masks, morphometric biomarkers (SAT, RF, and VI thickness) were computed using vertical profiles centered on the femur. Muscle quality was further characterized by extracting radiomic features from the RF region using PyRadiomics (van Griethuysen et al., 2017), including first-order statistics and gray-level co-occurrence matrix (GLCM) texture features (Paris and Mourtzakis, 2021). Segmentation performance was evaluated using the DSC. The agreement between ground truth and predicted biomarkers was assessed using Pearson correlation and the Concordance Correlation Coefficient (CCC).

## 3. Results and Discussion

**Segmentation and Anatomical Learning.** The 2D nnU-Net achieved high spatial overlap for skin and RF (DSC > 0.90) (Table 1). As shown in Fig. 1A, predicted contours closely follow annotated structures. The lower DSC for the femur (0.734) stems from its sparse point-based annotation; however, the network produces smoother, anatomically consistent masks that provide a stable reference for downstream analysis.

Table 1: Segmentation performance (DSC) and task-oriented validation (CCC, biomarker agreement) on the independent test set ($n = 71$).

| Structure | CV (DSC) | Test (DSC) | Biomarker CCC |
|---|---|---|---|
| Skin / SAT | $0.927 \pm 0.004$ | 0.934 | 0.990 |
| Femur / VI | $0.775 \pm 0.022$ | 0.734 | 0.978 |
| Rectus Femoris (RF) | $0.926 \pm 0.008$ | 0.935 | 0.895 |
| **Global Mean** | $0.876 \pm 0.009$ | 0.868 | $0.954 \pm 0.042$ |

**Clinical Biomarker Reliability.** Task-oriented validation provided preliminary evidence of the pipeline's feasibility for phenotyping. Morphometric biomarkers (SAT, RF, and VI thickness) achieved an excellent global mean CCC of $0.954 \pm 0.042$ (Table 1, Fig. 1B). Furthermore, muscle quality assessment via radiomics remained highly stable, with first-order features (mean, standard deviation, and kurtosis) and GLCM-based texture features yielding mean CCCs of 0.954 and 0.977, respectively.

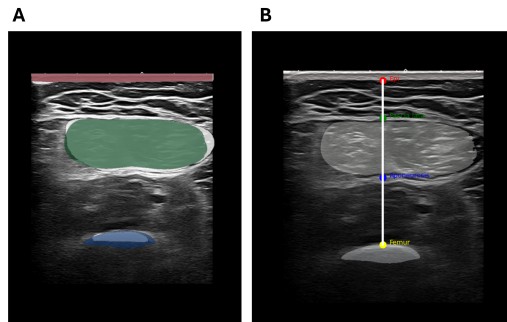

Figure 1: Results overview. (A) Predicted segmentations for skin (red), rectus femoris (green), and femur (blue) vs. expert ground truth (white contours). (B) Automatic biomarker extraction from predicted masks, enabling reliable thickness measurements for SAT, RF, and VI.

**Conclusion.** This study demonstrates that a self-configuring nnU-Net provides a promising pipeline for automated ultrasound phenotyping. The high agreement in both morphometric and radiomic biomarkers (CCC > 0.95) supports its potential clinical utility, addressing some limitations of manual assessment. Future work will focus on validating robustness across diverse populations and multi-center datasets.

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
