# OpenReview forum: "Task-Oriented Thigh Ultrasound Phenotyping via nnU-Net: From Segmentation to Biomarkers"
_MIDL.io/2026/Short_Papers — MIDL 2026 - Short Papers Poster_

### Official Review · Reviewer_8x6k · 2026-05-04
**Promising pipeline, but limited demonstrated impact**

**Rating:** 3
**Confidence:** 3

**Review:**

This short paper presents a practical pipeline for thigh ultrasound segmentation using a 2D nnU-Net, followed by extraction of morphometric biomarkers and radiomic features.

The paper is clearly written and addresses a relevant application in ultrasound-based body-composition assessment.
The strongest aspect is the task-oriented validation, where the authors evaluate agreement between biomarkers derived from manual and predicted segmentations instead of relying only on Dice scores.

The originality and significance are moderate. The paper does not propose a new method, which is acceptable for a short applied paper, but the contribution beyond applying nnU-Net to this dataset remains limited.

It is not yet clear what the reader should learn beyond the observation that nnU-Net performs well on this segmentation task.

Based on the presented example, the segmentation task appears relatively straightforward when training data are available, making the high biomarker agreement somewhat expected.

The work would be more compelling if the extracted biomarkers were used in a downstream clinical analysis, for example to distinguish cohorts, predict relevant outcomes, or demonstrate added value over simpler manual or landmark-based measurements.

Overall, the current evidence mainly supports technical feasibility and internal agreement between manual and automated biomarker extraction.

Additional points:
Several claims are slightly too strong given the current evidence. The results support good internal agreement between biomarkers extracted from manual and predicted masks, but they do not yet establish clinical utility, robustness, or superiority over manual assessment. The authors should temper the wording throughout the abstract and conclusion.
Claims that I would consider too strong or insufficiently supported:

- “confirming the reliability of this approach for high-throughput, automated ultrasound-based phenotyping in clinical research”
The study shows promising agreement on one internal independent test set, but reliability for high-throughput clinical research would require broader validation across scanners, operators, centers, and image-quality conditions.

- “demonstrating the reliability of the proposed approach for quantitative clinical analysis”
The paper demonstrates agreement with manual segmentation-derived biomarkers, but not yet clinical validity or clinical usefulness.

- “Task-oriented validation confirmed the pipeline’s robustness for phenotyping”
CCC-based agreement supports consistency with the manual reference, but robustness is not fully assessed. Robustness would require testing across acquisition settings, cohorts, operators, or external data.

- “overcoming the limitations of manual assessment”
This is not shown directly. The paper does not quantify manual inter-reader or intra-reader variability, nor does it compare the automated pipeline to the actual clinical measurement workflow.

- “supports its clinical utility”
The current results support technical feasibility and biomarker agreement, but clinical utility would require showing that the biomarkers are associated with relevant clinical outcomes, improve assessment, or reduce variability/time compared with current practice.

**Summary:**

This short paper presents an automated pipeline for thigh ultrasound analysis using a 2D nnU-Net to segment skin, rectus femoris, and femur, followed by extraction of biomarkers and radiomic features. The main contribution is the task-oriented evaluation. Instead of only reporting Dice scores, the authors assess agreement between biomarkers derived from manual and predicted segmentations. Reported results show high Dice scores for skin and rectus femoris, lower performance for femur, and high concordance for derived biomarkers.

**Strengths:**

•	The paper addresses a relevant clinical imaging problem: automated quantification of muscle and adipose tissue from ultrasound.

•	The task-oriented validation is a good choice and more meaningful than reporting segmentation metrics alone.

•	The use of an independent test set with patient-level separation is appropriate for a short paper.

•	The paper is clear, concise, and presents a practical pipeline that could be useful for larger body-composition studies.

**Weaknesses:**

•	The main limitation is not necessarily the lack of a new method. For a short applied paper, using a standard nnU-Net can be acceptable. However, the paper currently does not make sufficiently clear what the reader should learn beyond the fact that nnU-Net works well on this dataset.

•	The combination of thigh ultrasound segmentation, biomarker extraction, and task-oriented validation is reasonable, but the overall contribution remains somewhat incremental. The segmentation task appears relatively simple based on the presented example, at least when training data are available, and the high biomarker agreement is therefore not very surprising.

•	The task-oriented validation is a strength, but it only evaluates whether biomarkers from predicted segmentations correlate with biomarkers from manual segmentations. It does not yet show that these biomarkers are clinically meaningful, predictive, or useful for distinguishing relevant patient groups.

•	It is unclear why full segmentation is needed for the intended measurements. If the main biomarkers are thickness measurements, the authors should better justify why this pipeline is preferable to simpler manual, landmark-based, or semi-automatic clinical measurements.

**Justification Of Rating:**

The paper is clear and addresses a relevant application, and the task-oriented validation is appropriate. However, the contribution is limited beyond applying nnU-Net to this dataset, and the clinical relevance of the extracted biomarkers is not demonstrated. Several claims about robustness and clinical utility are stronger than the presented evidence supports.

---

### Decision · Program_Chairs · 2026-05-08

Accept (Poster)